# Detection of Selection Signatures in Anqing Six-End-White Pigs Based on Resequencing Data

**DOI:** 10.3390/genes13122310

**Published:** 2022-12-08

**Authors:** Yige Chen, Xudong Wu, Jinglin Wang, Yinhui Hou, Ying Liu, Bo Wang, Xiaojing Hu, Xianrui Zheng, Xiaodong Zhang, Yueyun Ding, Zongjun Yin

**Affiliations:** 1College of Animal Science and Technology, Anhui Agricultural University, Hefei 230036, China; 2Anhui Provincial Key Laboratory of Livestock and Poultry Product Safety Engineering, Institute of Animal Husbandry and Veterinary Medicine, Anhui Academy of Agricultural Sciences, Hefei 230031, China

**Keywords:** Anqing six-end-white pig, single nucleotide polymorphism, genetic diversity, positive selection, gene enrichment analyses

## Abstract

As a distinguished Chinese indigenous pig breed that exhibits disease resistance and high meat quality, the Anqing six-end-white (AQ) pig represents a valuable germplasm resource for improving the quality of the pig breeding industry. In this study, 24 AQ pigs that were distantly blood-related and 6 Asian Wild Boar (AWB) were selected for 10× deep-genome resequencing. The signatures of the selection were analyzed to explore the genetic basis of their germplasm characteristics and to identify excellent germplasm-related functional genes based on NGS data. A total of 49,289,052 SNPs and 6,186,123 indels were detected across the genome in 30 pigs. Most of the genetic variations were synonym mutations and existed in the intergenic region. We identified 275 selected regions (top 1%) harboring 85 genes by applying a crossover approach based on genetic differentiation (F_ST_) and polymorphism levels (π ratio). Some genes were found to be positively selected in AQ pigs’ breeding. The *SMPD4* and *DDX18* genes were involved in the immune response to pseudorabies virus (PRV) and porcine reproductive and respiratory syndrome virus (PRRSV). The *BCL6* and *P2RX6* genes were involved in biological regulation of immune T cells and phagocytes. The *SLC7A4* and *SPACA4* genes were related to reproductive performance. The *MSTN* and *HIF1A* genes were related to fat deposition and muscle development. Moreover, 138 overlapping regions were detected in selected regions and ROH islands of AQ pigs. Additionally, we found that the QTLs with the most overlapping regions were related to back fat thickness, meat color, pH value, fatty acid content, immune cells, parasitic immunity, and bacterial immunity. Based on functional enrichment analysis and QTLs mapping, we conducted further research on the molecular genetic basis of germplasm traits (disease resistance and excellent meat quality). These results are a reliable resource for conserving germplasm resources and exploiting molecular markers of AQ pigs.

## 1. Introduction

Wild boar and other large mammals were important prey for early hunter-gatherers. The domestication of livestock, including pigs, signified a major shift in socioeconomic transition in human history [1,2]. Extensive archaeology research has proved that pigs were first domesticated in the Near East around 10,000 years ago and then domesticated at multiple locations across Eurasia [3]. Domesticated individuals exhibit significant changes in physiology, morphology, and behavior compared with their wild ancestors. During their long history of evolution and breeding, pigs have been naturally and artificially selected to meet various needs for people all over the word. Hundreds of pig breeds have evolved with different breeding goals and selection systems. Most of these breeds exhibit special characteristics such as high fertility (Meishan pigs), hypoxic adaptation (Tibetan pigs), advantageous growth, and high feed conversion rates (Large White pigs) [4,5,6].

Modern molecular biology techniques are increasingly being applied to analyze the genetic structure and diversity of livestock. Twenty-five microsatellite markers were used to assess the genetic variability of the Uruguayan pig breed Pampa Rocha [7]. Sari et al. [8] used the mtDNA D-loop to obtain basic data on the genetic diversity of Aceh cattle and their association with Bhutanese, Chinese, and Indian cattle. With the advent of genome-wide dense single-nucleotide polymorphism (SNP) chips, genome-wide studies have emerged as a viable new method to investigate population history, genetic information, signatures of selection, and candidate genes of production traits in animal breeding. For example, Brito et al. [9] assessed the effective population size and population structure parameters of New Zealand multibreed sheep using an SNP chip with 14,845 samples; Wu et al. [10] described the population structures, genetic diversity, inbreeding coefficients, and selection signatures of Diannan small-ear (DSE) pigs in three different regions based on next-generation sequencing data. Genome-wide association analysis (GWAS) can identify significant SNPs or QTLs associated with pig production performance, such as growth and carcass and meat quality [11].

Artificial and natural selection have played a key role in the domestication of breeds and the improvement in livestock to adapt to their environment and needs [12]. It is well known that there is a close relationship between molecular and phenotypic changes. The process of adaptive evolution has developed in modern animal genomes, and many genetic footprints, such as signatures of selection and homozygosity runs, remain in the genome [13,14,15]. The advances and developments in genome sequencing and high-density chips help trace the history of modern animal domestication and the evidence of artificial and natural selection, and whole-genome-wide scanning has been widely used in animal husbandry.

In China, there are more than 80 indigenous pig breeds, constituting a valuable gene bank. The Anqing six-end white pig, an important local pig breed in the province of Anhui in China, has a strong physique, rough feeding resistance, strong stress resistance, and high reproduction rate, and is an ideal hybrid female parent for high-quality meat pigs [16]. Based on previous work, we investigated the runs of homozygosity (ROH) and copy number variations (CNVs) of AQ pigs, identifying a number of candidate genes associated with economic performance traits [13]. In this study, F_ST_ and π ratio analyses were used to detect selection signatures on genomic regions and explore the genetic basis of AQ pigs.

## 2. Materials and Methods

In this study, all animal trials and sample collection procedures were reviewed and approved by the Animal Welfare Committee of Anhui Agricultural University (permit number: AHAU20140215).

### 2.1. Sample Collection and Whole-Genome Resequencing

In this study, 30 unrelated porcine ear tissue samples were collected from AQ pigs (24 individuals from Taihu County, Anqing City, Anhui Province, China) and AWB (6 individuals from Mengla County, Xishuangbanna Dai Autonomous Prefecture, Yunnan Province, China). In a previous study, DNA samples were resequenced with an average depth of approximately 10× on an Illumina sequencing platform (Illumina, San Diego, CA, USA) at Genedenovo Biotechnology Co., Ltd. (Guangzhou, China). The sequence variants were called using the Genome Analysis Toolkit (version 4.0.2.1), and quality control standards were set as follows:-Window 4, filter “QD < 4.0 || FS > 60.0 || MQ < 40.0”, G_filter “GQ < 20” (QD: variant confidence/quality by depth; FS: Phred-scaled *p*-value using Fisher’s exact test to detect strand bias; MQ: RMS mapping quality; GQ: genotype quality) [13]. SNP genome coordinates were obtained from the Sus scrofa 11.1 porcine genome reference assembly (https://www.ncbi.nlm.nih.gov, accessed on 12 September 2022). Li, et al. [17] and Zhang et al. [18] described the analytical procedures of variant calling, SNP filtering, and annotation of the filtered SNP [18,19]. We conducted quality control of the SNPs to improve the accuracy of the selection signatures analysis. The filtering conditions were as follows: the minimum allele was less than 0.05, the SNP detection rate was less than 0.9, and marker loci with deletion rates over 20% were removed.

### 2.2. Genome-Wide Selective Sweeps Detection

The selection signatures of two pig breeds were identified with two different statistical methods: genetic differentiation (F_ST_) and pairwise nucleotide variation as a measure of variability (π ratio). In this study, we used sliding windows to scan the filtered SNPs of the whole genome and a 100 kb sliding window approach with a 10 kb step size was applied to calculate these statistics with PopGenome software (version 2.7.5) [19]. Based on the scan results, the overlapped window regions between F_ST_ and π ratio analyses with top 1% levels were considered as selection signatures, where were detected candidate genes in sweep regions.

### 2.3. Annotation of the Candidate Genes and Identifying QTL Overlapping with Selection Signatures and ROH Islands

To further analyze the functions of the identified genes, Gene Ontology (GO) and Kyoto Encyclopedia of Genes and Genomes (KEGG) analyses were performed using KOBAS (3.0) software (http://kobas.cbi.pku.edu.cn, accessed on 5 October 2022) [20]. Additionally, the specific selection direction of livestock increases the homozygosity in genomic regions, and the distribution of ROH islands across the genome shows that the genome has been subjected to selective pressure.

Therefore, we used the pig quantitative trait loci (QTL) database [21] (https://www.animal-genome.org/cgi-bin/QTLdb/SS/index, on SS11.1 in gff format, accessed on 27 October 2022) to annotate economic traits associated with the overlapping regions between sweeps selection and ROH islands.

## 3. Results

### 3.1. Whole-Genome Resequencing

In this study, a total of 935.04 Gb of raw data was obtained for the 30 swine genomes, which were submitted to NCBI under the accession number PRJNA699491; the average depth was 12.12X, and the summary statistics for the resequencing data were described (Appendix A). Among the 49,289,052 SNPs detected in all animals, 23,545,054 SNPs were detected in AQ pigs and 25,743,998 SNPs were detected in AWB. In addition, among the 6,186,123 indels detected in 30 pigs, 3,130,836 indels were detected in AQ pigs and 3,055,287 indels were detected in AWB. The number of transition SNPs in AQ pigs and AWB was much higher than the number of transversion SNPs, and the specific values of transition/transversion were 2.46 and 2.48, respectively (Appendix A). The SNP annotation results indicated that these variations were most abundant in the intergenic region and least in the exon region (Figure 1 and Figure 2).

### 3.2. Genome-Wide Selective Sweeps Detection

After variant calling and the subsequent stringent quality filtering, a total of 45,822,239 SNPs (23,185,417 were detected in AQ pigs and 22,636,822 were detected in AWB) with high quality were retained. In this study, a combination of F_ST_ and π ratio analyses was used for genome-wide selective sweeps detection, and the two thresholds were both at the top 1% level (F_ST_ was 0.46 and π ratio was 3.21) (Figure 3 and Figure 4). A total of 276 selective sweeps regions were identified in AQ pigs, satisfying extremely high F_ST_ values and significantly high π ratios (Figure 5). These regions were distributed in Sus scrofa chromosome (SSC) 1, 2, 6, 8, 13, 14, and 15. In these regions, 132 regions were observed on SSC15, whereas 4 regions were observed on SSC8. Furthermore, the 276 selective sweeps regions harbored 85 candidate genes.

### 3.3. Annotation of the Candidate Genes

We performed functional enrichment analysis on candidate genes that were in 132 selective sweeps regions. A total of 85 candidate genes were enriched in 2042 items, which were divided into 3 categories: biological process (BP), cellular component (CC), and molecular function (MF). The totals were 1558, 201, and 283, respectively (Figure 6, Appendix A). The enriched terms were mainly single-organism process, single-organism cellular process, viral replication complex, inclusion body, pattern binding, and polysaccharide binding. The KEGG pathway analysis revealed that candidate genes of the AQ pig genome selective sweeps regions were enriched in sphingolipid metabolism, renal cell carcinoma, inositol phosphate metabolism, and the Fc γ R-mediated phagocytosis pathway (Appendix A). An extensive and accurate search of the literature was performed, and several candidate genes were found to be associated with economic traits. The *SMPD4* gene has been reported as related to the neuronal invasion of pseudorabies virus (PRV) in pigs. The *DDX18* gene is involved in the inhibition replication of porcine reproductive and respiratory syndrome virus (PRRSV). The *BCL6* and *P2RX6* genes are involved in the biological regulation of immune T cells and phagocytes, respectively. The *SPACA4* gene is specifically expressed in the testis of male animals, and the *SLC7A4* gene is related to reproductive performance. The *MSTN* and *HIF1A* genes are related to fat deposition and muscle development in pigs.

### 3.4. QTL Identification

In this work, the pig quantitative trait loci (QTL) database was used to annotate economic traits related to the overlapped regions between sweeps selection and ROH islands. A total of 138 overlapped regions were identified in the whole genome, distributed in SSC13, 14, and 15 and harbored in 176 QTL loci. These QTL loci are classed into five types: reproduction QTL (*n* = 11), production QTL (*n* = 10), meat and carcass QTL (*n* = 125), health QTL (*n* = 24), and exterior QTL (*n* = 6); the number of QTL relating to meat and carcass is especially greater than the others, accounting for a proportion of 71%. The mapping results of QTLs are shown in Appendix A, such as QTL_ID.21409, QTL_ID.21407, and QTL_ID.2126, which were related to back fat thickness traits; QTL_ID.38113, QTL_ID.38107, and QTL_ID.21720, which were related to shear force traits; and QTL_ID.1144, QTL_ID.8810, and QTL_ID.5952, which were related to body mass traits.

## 4. Discussion

The pig industry is the most important in worldwide agriculture farming, and various native pig breeds constitute a valuable germplasm bank [22]. The generation and stable inheritance of new genetic variations are the bases of species evolution, and genetic variation detection is useful for understanding the processes that underlie the diversification of traits [23]. Currently, the assembly of a Duroc pig whole-genome sequencing (*Sus scrofa* 11.1) provides a good reference to explore the special trace in pig genomes [24]. Whole-genome resequencing is different from high-density chips, which have the obvious advantage of finding new nucleotide mutations and can indicate whole genetic marker information. In this study, we obtained the 935.04 G resequencing data of AQ pigs and AWB genome and called 45,822,239 SNPs by aligning with swine reference genomes; the alignment rate was 90.25%, indicating that the amount of data and sequencing quality obtained in this sequencing meet the requirements of further genomic research.

Selection signal refers to the genetic imprint in the genome of pressure under natural or artificial selection, which is characterized by the decrease in genetic diversity and the increase in chromosome linkage disequilibrium at the selected locus [25]. After the domestication of wild boars, domestic pigs have been subjected to a strong selective pressure to develop the desired phenotypes. The genomic regions and related genes involved in domestic pig domestication and breed selection can be obtained by comparing the genomic genetic information of domestic swine and wild boars. F_ST_ analysis is the preferred method to detect differences in allele frequencies between populations that are involved in local adaptation based on the detection of ‘outlier’ values [26]. π ratio analysis is also a common detection method used to focus on the polymorphisms of a specific population based on gene heterozygosity, and a larger π ratio indicates a higher polymorphism [27]. These methods have been widely used in the selection signal detection of livestock [28,29]. In Zhang’s study, F_ST_ and π ratio analyses were combined; used to define the genomic selection region of the Wannan black pig; and found selected genes associated with growth, immunity, and digestive functions [30]. In this study, we used the genome information of AWB as the reference, and the overlapped window regions between F_ST_ and π ratio analyses with the top 1% level were considered as selection signatures for AQ pigs.

In this study, a total of 275 selected regions were found, and the largest number of selected regions were on SSC 15. There were 85 candidate genes identified in the selected regions, and some candidate genes were found that were related to production traits. The *MSTN* gene is associated with meat quality, which was screened from the selected region on SSC 15 (94.62–94.72 Mb). Two mutation loci of the *MSTN* gene in Norwegian White sheep were significantly correlated with fat content, and the c. 960delG locus showed dominance effects [31]. Stinckens et al. [32] found that the *MSTN* gene may be a candidate gene for muscular hypertrophy in pigs. Through the GWAS analysis result, Fowler et al. [33] found two QTLs related to the back fat thickness trait of Large White pigs in adjacent locations of this selected region, indicating that this selected region may be related to the back fat thickness trait of AQ pigs. The expression of the *BCL6* gene is a necessary condition for the cell differentiation of immune T cells, and participates in the immune response of T cells and B cells [34]. Macrophages are immune cells that regulate the maintenance of tissue homeostasis, host defense during pathogen infection, and repair tissue in response to tissue injury [35]. P2X receptor proteins are implicated in the activation of immune cells. The *P2RX6* gene expression was lower than that of other P2X receptor families in healthy macrophages [36]. A recent report indicated that ATP-P2RX6 might modulate the Ca^2+^-mediated p-ERK1/2/MMP9 signaling to increase the RCC cells’ migration and invasion [37]. Porcine reproductive and respiratory syndrome virus (PRRSV) and porcine pseudorabies virus (PRV) are major infectious diseases that affect the development of the pig industry; once viruses have occurred in pig farms, they cause serious economic losses [38,39]. Here, two genes (DEAD-box helicase 18 (*DDX18*) and sphingomyelin phosphodiesterase 4 (*SMPD4*)) were related to PRRSV and PRV biological activities in AQ pigs, which were screened from the selected region. *DDX18* plays an important role in the replication of viral RNA and hosts innate immunity responses.

Jin et al. [40] proved that *DDX18* participates in PRRSV viral replication and *DDX18* interacts with both nsp2 and nsp10 of PRRSV by co-immunoprecipitation (Co-IP). PRV is an α-herpesvirus that requires the viral protein Us9 to sort virus particles into axons and facilitate neuronal spread. *SMPD4,* associated with Us9, is a negative regulator of PRV sorting into axons and neuronal spread. A potential antiviral function, the expression of the *SMPD4* protein was different between uninfected and infected neuronal cell lines [41].

Pavlidis et al. [42] pointed out that the functional enrichment annotation of selective sweeps genes did not perform well. The functional enrichment analysis of genes usually only focuses on biological information, and it is difficult to explain the intrinsic relationship between genes and complex quantitative traits. QTL mapping can fully consider genetic effects, interactions of molecular markers, and the influence of fixed and random effects on phenotype [43]. In this study, the pig QTL database was used to annotate economic traits related to the overlapped regions between sweep selection and ROH islands. Through the overlapped results, most overlapped regions related with meat and carcass QTL and health QTL. Based on previous research, AQ pigs exhibited excellent adaptive capacity and good meat quality [44]. During the breeding process, AQ pigs were fed with green forage and lived in dim and humid pigsties, which aroused a positive selection pressure for disease resistance. Both the functional annotation of candidate genes and QTL overlapping results are related to AQ pig characteristics.

## 5. Conclusions

In summary, we used the 10× resequencing method to detect the heritable variation information of AQ pigs and AWB. F_ST_ and π ratio analyses were used for genome-wide selective sweeps detection. Some selection signatures might be functionally associated with the meat quality and disease resistance of AQ pigs. The results help provide valuable genetic basis for AQ pig breeding.

## Figures and Tables

**Figure 1 genes-13-02310-f001:**
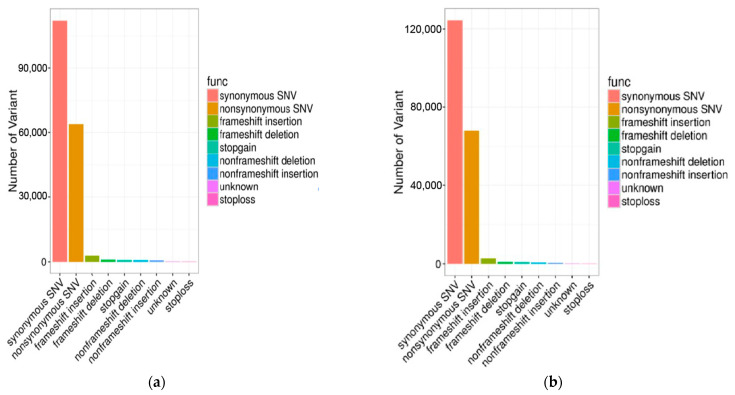
The functional annotation statistical map of genetic variation in Anqing six-end-white pigs (**a**) and Asian Wild Boar (**b**).

**Figure 2 genes-13-02310-f002:**
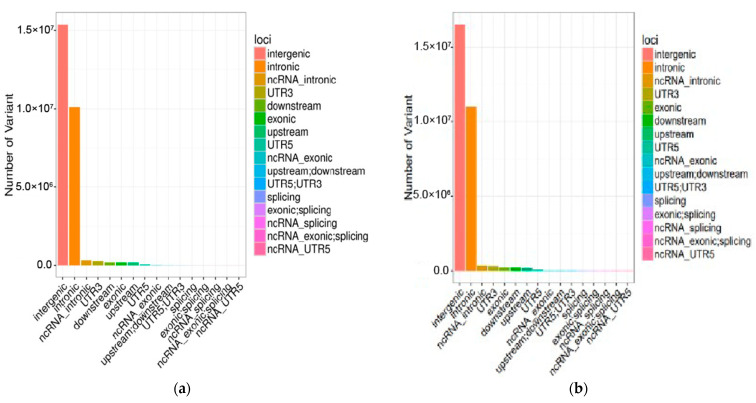
The statistical map of genetic variation locations of Anqing six-end-white pigs (**a**) and Asian Wild Boar (**b**).

**Figure 3 genes-13-02310-f003:**
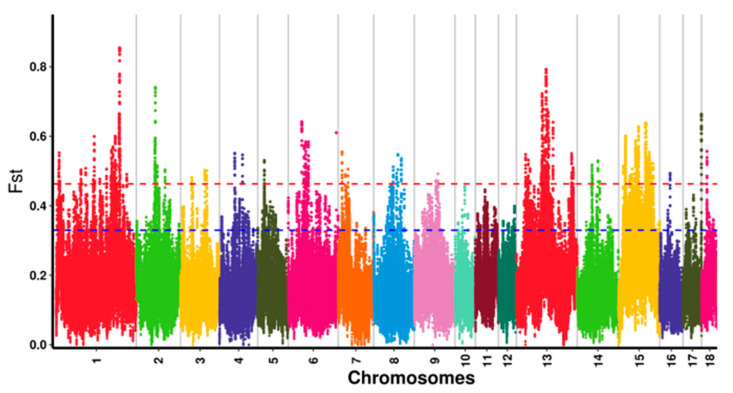
Distribution of F_ST_ values among Anqing six-end-white pig autosomal chromosomes. Note: The red line represents the 0.01 level, and the blue line represents the 0.05 level. The different colors indicate that the SNPs are located on different chromosomes.

**Figure 4 genes-13-02310-f004:**
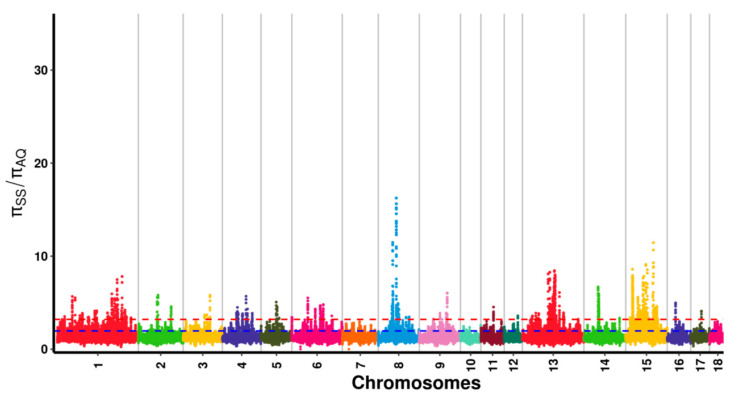
Distribution of π ratio among Anqing six-end-white pig autosomal chromosomes. Note: The red line represents the 0.01 level., and the blue line represents the 0.05 level. The different colors indicate that the SNPs are located on different chromosomes.

**Figure 5 genes-13-02310-f005:**
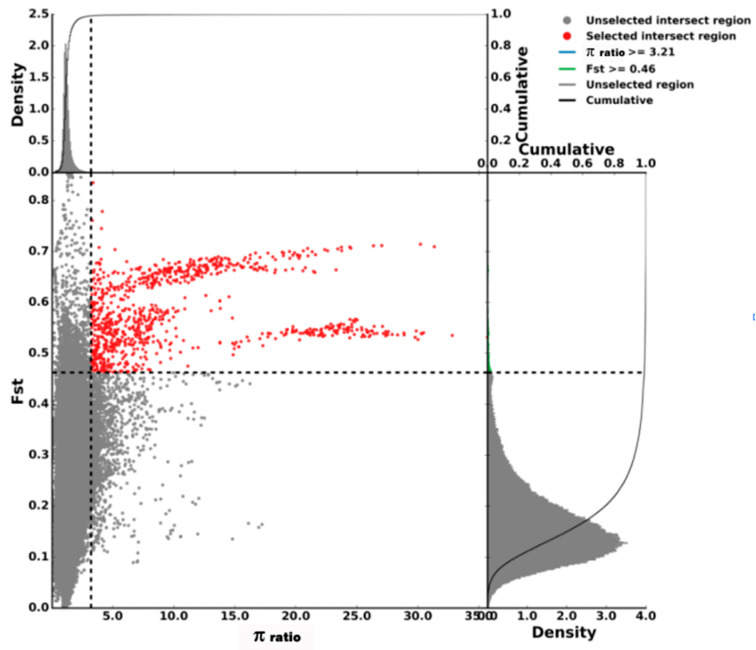
Selective sweeps regions identification of Anqing six-end-white pigs. Note: The final selection regions were based on two statistics. Points located to the right of the vertical dashed lines (corresponding to 1% right tails of the empirical π ratio distribution, where the π ratio was 3.21) and above the horizontal dashed line (1% above tail of the empirical F_ST_ distribution, where Fst was 0.46) were identified as selected regions for Anqing six-end-white pigs (red points).

**Figure 6 genes-13-02310-f006:**
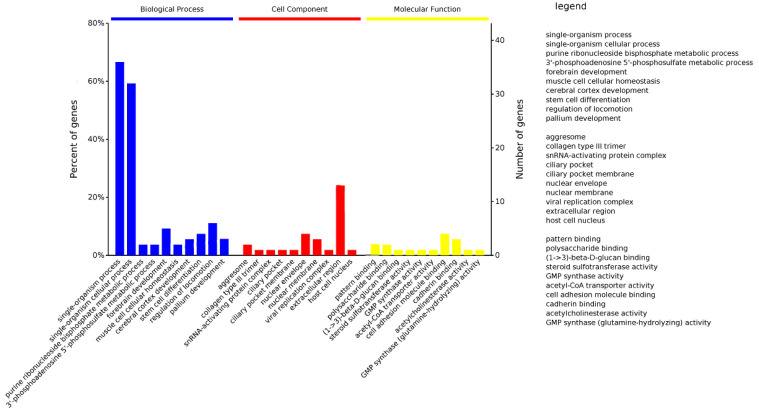
GO enrichment analysis of genes in selected regions of Anqing six-end-white pigs.

## Data Availability

The BioProject number of the sequencing data information about Anqing Six-end-white pigs and Asian wild boars are PRJNA699491 in the NCBI Sequence Read Archive.

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
