# Peer review of "Detection of Selection Signatures in Anqing Six-End-White Pigs Based on Resequencing Data"

_genes, 2022, doi:10.3390/genes13122310_

Round 1

Reviewer 1 Report

This article provides new information for genomic selection signature detection between Anqing six-end-white pig and Asian wild Boar. The purpose of the study is very interesting and important data also have been obtained. However, there are still some minor shortcoming could to be improved.

1.     In part 2.1, 2.2, please provide the version of Genome analysis Toolkit and PopGenome software;

2.     The names of pig breeds in Tables and Figures should be used the full name.

3.     In Figure 3 and Figure 4, the pig breed should be in the figure note. What the red line represent?

4.     The Figure 6 should be shown GO enrichment analysis of AQ pigs.

5.     The gene names should be in italic format.

Reviewer 2 Report

The manuscript investigated the different patterns of runs of homozygosity in AWBs and AQ pigs, and the authors highlighted the differences in ROH numbers, ROH rates and FROH values between AWBs and AQ pigs. The study design is quite straightforward, and the results are clearly described but there are several potential weaknesses.

1.    There are more than 2 studies to support the research for AQ pigs, all of them should be cited:

Hu H., Wu C., Ding Y., Zhang X., Yang M., Wen A. & Yin Z. (2019) Comparative analysis of meat sensory quality, antioxidant status, growth hormone and orexin between Anqingliubai and Yorkshire pigs. Journal of Applied Animal Research;

Guo L., Sun H., Zhao Q., Xu Z., Zhang Z., Liu D., Qadri Q., Ma P., Wang Q. & Pan Y. (2021) Positive selection signatures in Anqing six-end-white pig population based on reduced-representation genome sequencing data. Animal Genetics.

2.    Line56: there are more than 2 studies to explore the genetic characteristics of eastern China indigenous pigs, why only choose to mention DSE pig, which is the breed from Yunnan province?

3.    Please mention Anqing six-end-white pigs in the title as previous researches, but not only mention eastern China indigenous pigs.

4.    Line 71: please delete the needless space, check around the whole manuscript.

5.    Line 84: why did the authors use the AWB from Yunnan province, far from Anhui province. There were some WGS datasets of AWB from Jiangxi province publicly available, which were better to be used in this study.

6.    Line 109: FST should be correctly written.

7.    Line 116: the QTL database should be cited and the release version should be mentioned:

Hu Z, Park C, and Reecy J (2022). Bringing the Animal QTLdb and CorrDB into the future: meeting new challenges and providing updated services. Nucleic Acids Research

8.    Please provide the abbreviations at first use (e.g., AWB in Line 69 and SSC in Line 106).

Table and Figure:

Table 1: should be included in the supplemental materials;

Figure: Please, the font and size letters must be checked. There are different fonts and sizes used.

Round 2

Reviewer 2 Report

The authors have revised all my concerns.